

# Simultaneous assessment of oxygen and nitrate-based net community production in a temperate shelf sea from a single ocean glider

Tom Hull[1,2], Naomi Greenwood[1,2], Antony Birchill[3,4], Alexander Beaton[4], Matthew Palmer[4], and Jan Kaiser[1]

[1]Centre for Ocean and Atmospheric Sciences, University of East Anglia, Norwich, UK
[2]Centre for Environment, Fisheries and Aquaculture Science, Lowestoft, UK
[3]School of Geography, Earth and Environmental Sciences, University of Plymouth, Plymouth, UK
[4]National Oceanography Centre, Southampton, UK

**Correspondence:** Tom Hull (tom.hull@cefas.co.uk)

**Abstract.** The continental shelf seas are important at a global scale for ecosystem services. These highly dynamic regions are under a wide range of stresses and as such future management requires appropriate monitoring measures. A key metric to understanding and predicting future change are the rates of biological productivity. We present here the use of a single autonomous underwater glider with oxygen ($O_2$) and total oxidised nitrogen ($NO_x^- = NO_3^- + NO_2^-$) sensors during a spring bloom as part of a 2019 pilot autonomous shelf sea monitoring study. We find exceptionally high rates of net community production using both $O_2$ and $NO_x^-$ water column inventory changes, corrected for air-sea gas exchange in case of $O_2$. We compare these rates with 2007 and 2008 mooring observations finding similar rates of $NO_x^-$ consumption. With these complementary methods we determine the $O_2 : N$ amount ratio of the newly produced organic matter ($7.8 \pm 0.4$) and the overall $O_2 : N$ ratio for the total water column ($5.7 \pm 0.4$). The former is close to the canonical Redfield $O_2 : N$ ratio of $8.6 \pm 1.0$, whereas the latter may be explained by a combination of new organic matter production and preferential remineralisation of more reduced organic matter at a higher $O_2 : N$ ratio below the euphotic zone.

## 1 Introduction

The coastal shelf seas are a vitally important human resource for numerous ecosystem services, including food, carbon storage, biodiversity, energy and livelihoods (Halpern et al., 2015). They have a disproportionately large impact, relative to their surface area, on global carbon cycling (Thomas, 2004) as shelf seas provide 10-30 % of all marine primary production while comprising less than 10 % of the ocean surface (Harris et al., 2014; Sharples et al., 2019). Given their global role in carbon cycling, understanding the mechanisms driving primary production is vital for predicting how shelf systems will respond to and influence climate change (Palevsky et al., 2013; Legge et al., 2020).





## 1.1 Carbon cycling in the Central North Sea

The North Sea is a semi-enclosed region of the Northwest European continental shelf between the island of Great Britain in the west and Norway, Denmark, Germany and the Netherlands in the east and south. The northern boundary is facing the North Atlantic. It can be divided into two energetically distinct regimes, a seasonally stratifying northern part, and a fully mixed southern and coastal part (Emeis et al., 2015). The shallower fully mixed region is believed to act as a net sink for $CO_2$ during the spring bloom periods, and a source of $CO_2$ during the rest of the year (Emeis et al., 2015; Hull et al., 2016). Thomas

et al. (2005b) suggests the seasonally stratified regions are a sink for carbon throughout the year. A more recent synthesis of Kitidis et al. (2019) indicates that the seasonally stratified regions were a source of $CO_2$ during December and January in 2015. Dissolved inorganic carbon from organic matter remineralisation (respiration) accumulates in the waters below the seasonal thermocline throughout the spring and summer. In winter the deepening of the thermocline due to increased wind stress and reduction of surface heat fluxes partly remixes bottom waters high in dissolved inorganic carbon back into the surface layer.

However, the bulk of water entering the North Sea does so near the surface and leaves the shelf at depth. This downwelling circulation results in a substantial transport below the pycnocline from the North Sea into the Norwegian Trench (13 %) and the subsurface North Atlantic (6 %) prior to seasonal remixing (Holt et al., 2009). The majority of the atmospheric carbon fixed during annual primary production is exported to the North Atlantic through the shelf sea carbon pump (Thomas, 2004), with a very small fraction, less than 1 %, actually being buried in the North Sea sediments and 90 % exported to the North

Atlantic (Thomas et al., 2005a). Overall, the majority of pelagic North Sea dissolved inorganic carbon both originated from and is exported back to the Atlantic via cross-shelf exchange and plays a limited role in the net shelf carbon cycle (Legge et al., 2020). In additional to seasonal cycling, carbon cycling in the central North Sea may be influenced by periodic flushing over longer periods (Humphreys et al., 2018; Chaichana et al., 2019).

## 1.2 Hydrography and biogeochemistry at Dogger Bank

The Dogger Bank is a large sandbank within the central North Sea (fig. 1). Comprising predominantly fine sand and mud, this bank rises 20 m above the surrounding sea floor and is 15 m below sea level in its shallowest parts. Tidal forcings are dominated by the M2 constituent, with mean tidal excursion in the region of 1.5 km (fig. 1). The Dogger Bank sits south of a tidal front, which is located in an arc across the central North Sea from Yorkshire in the United Kingdom to the Frisian coast in the Netherlands.This hydrological divide separates well-mixed waters to the south from the seasonally stratified waters of

the central North Sea to the north (Emeis et al., 2015). The combination of tidal, wind and buoyancy forcing tends to develop a general anti-clockwise circulation within the North Sea (Mathis et al., 2015). Variability in the circulation is determined mainly by variation in local winds, which are predominantly from the west. Prevailing currents move from the west along the tidal front towards the Dutch coast. These currents are supplied with water from the Scottish coastal current, itself primarily fed from the North Atlantic inflow through the Orkney-Shetland gap (Hill et al., 2008). The Dogger Bank experiences minimal riverine

influence, such that maximal nutrient concentrations typically reflect those of the in-flowing Atlantic waters (Greenwood et al.,



2010). The changes in N:P ratio observed in more coastal regions of the North Sea are not seen at Dogger Bank and nitrogen remains the primary limiting nutrient (Burson et al., 2016).

The shallow waters above Dogger Bank are not seasonally stratified, as wind-stress and tidal currents regularly homogenise the shallowest areas, which replenishes nutrients into the photic zone throughout the year (Riegman et al., 1990). In this paper we focus on the region north of the Dogger Bank where mixed waters converge with the seasonally thermally stratified waters. The exchange of nutrients and phytoplankton at this transition promotes enhanced primary production along the northern edge of the bank, which acts as a hotspot for marine life and supports several important fisheries (Plumeridge and Roberts, 2017). Since 2017, the bank has been designated as a Special Area of Conservation (SAC) by the UK government, signifying it as a region containing habitats and species which are considered most in need of conservation and legal protection. The area on and around Dogger Bank is thought to exhibit year-round phytoplankton production (Nielsen et al., 1993). Due to the shallow depth of Dogger Bank, the spring phytoplankton bloom can be initiated there months before stratification triggers the bloom within the more northern stratified regions of the North Sea (Nielsen et al., 1993). The region north of Dogger Bank typically sees a spring phytoplankton bloom commencing at the start of April.

### 1.3 The need for improved monitoring

The North Sea is one of the most well studied marine environments. However, despite the availability of large volumes of data, making robust statements on the long-term changes, natural variability or predicted process responses to external forcing continues to be challenging (Emeis et al., 2015). As is typical for shelf regions, the North Sea displays high temporal and spatial variability, which makes the analysis of trends more challenging compared with the open ocean (Bozec et al., 2006; Bauer et al., 2013). From a carbon system point of view, it remains uncertain whether the shelf sea carbon sink will strengthen or weaken in the near future as this complex system is being influenced by multiple, interacting factors (Legge et al., 2020; Macovei et al., 2021).

The North Sea is changing and the processes controlling carbon export and the oxygen dynamics are highly complex and interconnected (Wakelin et al., 2020). Key among those processes is marine productivity. Estimates of net community production (NCP) can quantify the uptake of $CO_2$ from the atmosphere into the sea (Alkire et al., 2012; Hull et al., 2016; Plant et al., 2016). Changes to community productivity can affect how organic material is exported to and respired within the non-ventilated waters below the thermocline (Große et al., 2016). NCP is therefore a useful metric for understanding carbon cycling and an important control on bottom water oxygen concentration.

Seasonal low oxygen concentrations have been observed in the bottom layer of the seasonally stratified waters north of the Dogger Bank, and various monitoring studies have shown a general decline since at least 1990 (Greenwood et al., 2010; Queste et al., 2013; Große et al., 2016). While concentrations are still well above $2\,\mathrm{mg\,L^{-1}}$ ($63\,\mathrm{mmol\,m^{-3}}$), which is often considered to be the threshold of hypoxia (Diaz and Rosenberg, 2008), oxygen concentrations have been predicted to be approaching the low-oxygen threshold of $6\,\mathrm{mg\,L^{-1}}$ ($188\,\mathrm{mmol\,m^{-3}}$) used by the OSPAR Commission, and as such are a growing concern for the future of the region (Ciavatta et al., 2016; Mahaffey et al., 2020). An improved understanding, led by enhanced monitoring, is essential to enable prediction and the mitigation of deleterious effects.

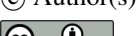



At present, resolving carbon cycling and oxygen dynamics in shelf sea regions remain a challenge for numerical biogeochemical models (Bauer et al., 2013; Emeis et al., 2015). Further development of model parametrisations for these processes, in addition to validation of the model outputs over appropriate spatial and temporal scales, is required. Discrete samples from shipborne hydrographic surveys can present a highly detailed synoptic view of various parameters, but these approaches will typically alias much of the temporal variability, including inter-annual variation. Rovelli et al. (2016) argued that in the absence

of targeted long-term studies focusing on oxygen and carbon dynamics across the whole water column, it is not possible to determine the long-term fate of bottom mixed layer oxygen concentrations. Sustained observations are therefore required to detect, understand and predict the conditions controlling the depletion of oxygen in UK marine waters (Mahaffey et al., 2020).

## 1.4    The AlterEco project

The glider-based oxygen and total oxidised nitrogen ($NO_x{}^- = NO_3{}^- + NO_2{}^-$) concentration measurements presented within

this paper form part of the larger AlterEco ("Alternative framework to assess marine ecosystem functioning in shelf seas"; https://altereco.ac.uk) project. AlterEco is a pilot study of a novel monitoring framework to deliver improved spatiotemporal understanding of key shelf sea ecosystem drivers though the use of autonomous systems, primarily underwater gliders, as part of the NERC Marine Integrated Autonomous Observing Systems programme. Buoyancy gliders are capable of directed, long-term continuous monitoring over a variable spatial area while covering the full water column within a shelf sea (Alkire

et al., 2012; Wihsgott et al., 2019). Gliders present an opportunity to combine some of best aspects of monitoring buoys and classical shipborne CTD surveys. With high resolution and high quality glider based measurements, we can simultaneously resolve nutrient and oxygen fluxes, and determine the net community production associated with the spring bloom at a finer scale than previously determined.

## 2    Data collection and method

## 2.1    Study area

Observations were made along an east-west transect (fig. 1) north of the Dogger Bank between 2019-03-29 and 2019-04-25 with an underwater glider (Seaglider sg602 "Scapa"). This period was the seventh group of AlterEco glider deployments and is therefore referred to by the mission code "AE7". The Seaglider took 5 days to move from one end of the transect to the other and completed six transits. The glider was fitted with a standard non-pumped Seabird CT sail, an Aanderaa 4831 oxygen

optode (Bittig et al., 2018) and a NOC Lab-on-a-Chip (LoC) microfluidic $NO_x{}^-$ analyser (Beaton et al., 2012; Vincent et al., 2018).

    The Seaglider conductivity, temperature (CT) and oxygen data were processed and quality-assured using the UEA Seaglider toolbox (Queste, 2013). This comprised optimising the Seaglider flight parameters to determine speed though water, which was used to apply a cell thermal mass correction for the CT sensor. This is an iterative process as determining glider flight

requires determining the water density, which is calculated from the measured salinity (Garau et al., 2011). Some salinity

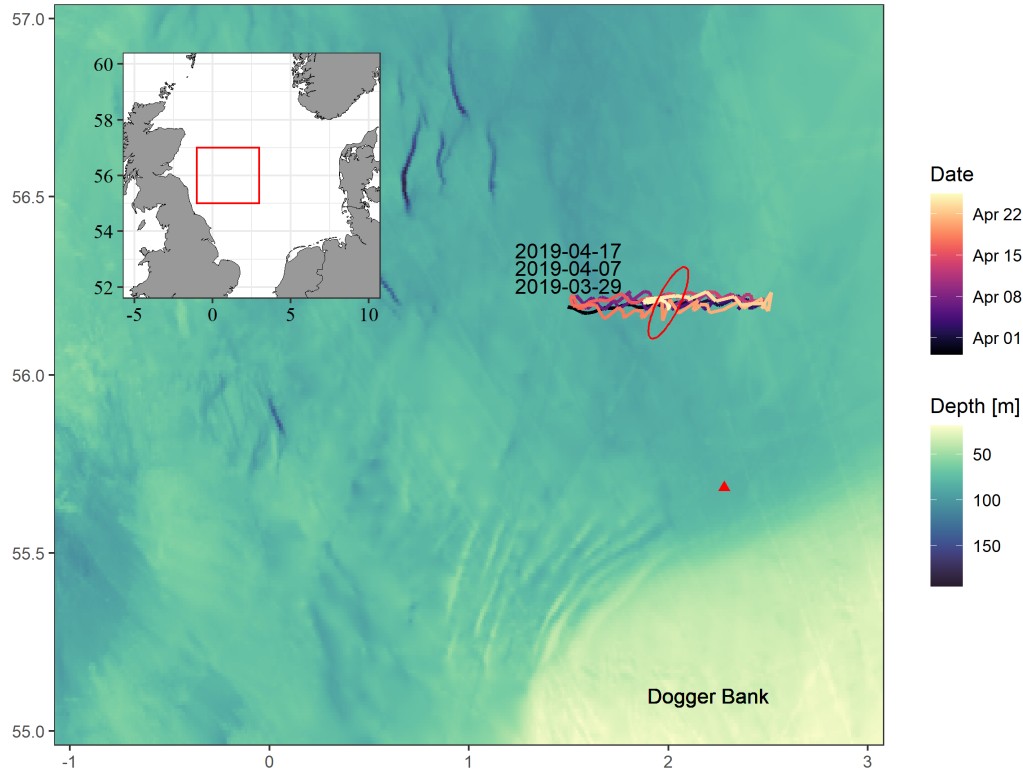

**Figure 1.** Seaglider path for the AlterEco AE7 2019 mission and the surrounding study area. Timestamps indicate when the glider returned to the western side of the transect. Mean tidal ellipse shown in red. Red triangle marks the location of the Cefas North Dogger SmartBuoy, present between 2007 and 2008. Bathymetry data from GEBCO 2019 on a 15 arc-second grid.

spikes still persist after performing the cell thermal mass correction due to strong vertical temperature gradients towards the end of the deployment ($>0.8\,°C\,m^{-1}$); these were manually quality-flagged (marked as bad) on a per-dive basis and not used in the analysis. As the Seaglider CT sensor relies on the forward movement of the vehicle to push water through the measurement cell, salinity and temperature values were also removed from the analysis where the glider speed was considered too slow to
provide adequate flow through the cell ($<0.1\,m\,s^{-1}$).

For parts of the following analysis, we treat the water column as a three layer system, with a surface (SML) and bottom mixed layer (BML) separated by a transitional thermocline. Both the SML and BML are considered to be homogeneous, maintained by wind and tidal mixing respectively. The extent of these layers is typically calculated from the vertical profile of temperature (or density). Using the depth at which the temperature is sufficiently different from a reference depth (within the
homogeneous layer) as the base of the layer. A reference depth of 10 m, which is commonly used for determining the surface mixed layer depth ($z_{mix}$) in open ocean mass balance studies (de Boyer Montégut, 2004; Kara et al., 2000), is inappropriately deep for our data (fig. 2 **b**). Data within the top 3 m is of poor quality due to the Seaglider surface manoeuvres and wave action.



A reference depth of 5 m is therefore chosen as deep enough to avoid biases from poor surface data quality while shallow enough to determine the air-sea oxygen gradient (see section A1). We adopt a temperature based threshold for $z_{mix}$ of 0.2 ºC

(as per Castro-Morales and Kaiser (2012)) which visually agrees with oxygen profiles (fig. 2 **b-d** and fig. A1). $z_{mix}$ is therefore defined here as shallowest depth at which the temperature differs (warmer) from that at 5 m by more than $0.2\,°C$. We reverse the procedure for the bottom mixed layer depth and use the deepest temperature recorded the reference threshold. The choice of SML or BML definition has minimal impact on our analysis as we do not use the SML as our integration depth for the $NO_x^-$ and $O_2$ mass balance.

## 2.2   $NO_x^-$ observations

The LoC $NO_x^-$ sensor is a miniaturised wet-chemical colorimetric analyser offering high precision in-situ total oxidised nitrogen measurement (Beaton et al., 2012). The analyser takes approximately 20 s to collect a sample, and a further 7.5 min to perform an analysis.The concentration of the on-board artificial seawater nutrient standard was determined by laboratory continuous gas segmented flow analysis (Beaton et al., 2012; Vincent et al., 2018). Accuracy was determined with reference to

two autoclaved natural seawater standards (KANSO Co. LTD, JAPAN) and summarised in table A1. $NO_x^-$ values below the limit of detection are treated as $0\,mmol\,m^{-3}$) in our calculations.

The limit of detection is $0.025\,mmol\,m^{-3}$. Vincent et al. (2018) demonstrated excellent stability and agreement between data generated LoC analysers deployed on an ocean glider and samples simultaneously collected by traditional shipboard techniques and analysed via standard gas segmented flow analysis. This Celtic Sea study region of Vincent et al. (2018) represents a similar shelf sea environment which exhibits a similar range of $NO_x^-$ concentrations to the North Sea.

As the maximum depth along the AlterEco transect is less than 85 m, the glider was configured to perform periodic loiter dives to enable multiple samples to be collected and so maximise the vertical resolution of the $NO_x^-$ samples (Vincent et al., 2018). With the standard Seaglider flight mode the glider pitch and buoyancy is adjusted to maintain a uniform speed during both the descent and ascent phase. A loiter dive consists of a standard descent phase, during which a blank and standard

measurement are performed, followed by a deliberately slow ascent which provides time for the $NO_x^-$ sensor to take more samples ($n$ between 3 and 14). Along the transect standard dives took $(30\pm3)\,min$, while loiter dives took $(110\pm5)\,min$. The reduced speed of the glider during loiter ascent results in inadequate flushing of the CT cell, with impaired salinity data quality. We therefore extrapolate the salinity from the descent to the ascent $NO_x^-$ data.

Photosynthetic fixation of atmospheric carbon into new organic material by phytoplankton is associated with a corresponding

consumption of inorganic nutrients (Anderson and Sarmiento, 1994). Where $NO_x^-$ is the main source of nitrogen, the $NO_x^-$ uptake represents net community production (Plant et al., 2016). As the number and position in the water column of the samples is variable, the inventories are calculated on a per dive basis after linear interpolation onto a 1 m vertical grid. the sum of which is then calculated to provide a full water column inventory. The standard uncertainty of the inventory is calculated from the squared sum of the relative $NO_x^-$ concentration uncertainties ($\pm0.13\,\%$). and a $0.5\,m$ uncertainty in water column height.

We exclude three dives were we are unable to calculate a full water column inventory as either the vertical spacing between samples near the thermocline was too large ($>10\,m$) or if samples are unavailable from each of the three vertical layers (fig. 2





**d**). For direct comparison with the Vincent et al. (2018) central Celtic Sea study we also calculate $NO_x^-$ drawdown within the upper section of the water column down to a $40\,\mathrm{m}$ depth. This coincides with the mean euphotic depth for our study region in April 2018 as observed by another AlterEco Seaglider ($z_{\mathrm{EU}} = (40 \pm 8)\,\mathrm{m}$) (Hull and Kaiser, 2020). The euphotic zone depth
here being defined as the depth to which photosynthetically active radiation (PAR) falls to 1% of the level measured at the surface.

### 2.3   $O_2$ observations

During typical (non-loiter) dives the Seaglider's vertical speed was typically $0.1\,\mathrm{m\,s^{-1}}$ and the oxygen sampling period was $10\,\mathrm{s}$. The optode was multi-point calibrated (10 concentrations at 4 temperatures each, 40 points total) by Aanderaa in January
2019 and referenced to in situ Winkler titrations taken during deployment and recovery. We adopted the recommendation of Bittig et al. (2018) that in the absence of reference data spanning a wide range of oxygen concentrations a correction factor (gain) should be applied to calibrate the oxygen concentration. i.e. where $C_{\mathrm{corrected}} = a + Cb$, $a = 0$ and $b > 1$. This is in contrast to an offset, or a combination of offset and factor, or a correction applied to the raw phase reading of the optode (Uchida et al., 2008). Reference to the discrete samples remained challenging as the deployment location was within a region
of significant horizontal heterogeneity. The reference location was as close as possible to the study region, but was legally constrained to within $60\,\mathrm{nm}$ from the coast due to the small coastal vessel used. Drift in the optode response to oxygen was assumed to be linear with respect to time and corrected based on the deployment and recovery samples ($-0.025\,\%\,\mathrm{d^{-1}}$) (Tengberg and Hovdenes, 2014; Bittig et al., 2018).

   Biological oxygen production can be used as a proxy for net community production (NCP), with a net production of oxygen
indicative of a net autotrophic system (Alkire et al., 2014; Hull et al., 2016). We implement an oxygen mass balance model to separate the physically and biologically driven components of the observed oxygen inventory changes. As with the $NO_x^-$ mass balance we calculate values for the euphotic zone ($z_I = 40\,\mathrm{m}$) and the full water column ($z_I = 82\,\mathrm{m}$). Full details of this model and its implementation are found in the supplementary section A1.

### 2.4   Supporting data

Local wind speed and atmospheric pressure were obtained from ECMWF ERA5 reanalysis (Copernicus, 2018). These hourly data were matched to the spatiotemporally closest Seaglider dive. For ocean currents at the AlterEco site we use the UK Met Office Operational Suite, Atlantic Margin Model (FOAM) 1.5 km configuration (run 2020-05-18) (Graham et al., 2018). Bathymetry was taken from the GEBCO 2019, on a 15 arc-second grid (GEBCO, 2019). Between 2007 and 2008 an autonomous monitoring buoy (SmartBuoy, Greenwood (2016)) was located to the south of the Seaglider transect (fig. 1). This
platform collected near surface ($0.5\,\mathrm{m}$) $O_2$ and $NO_x^-$ concentration measurements using similar sensors to the Seaglider. Full details of this platform are provided in the supplementary material A2.





## 3 Results

### 3.1 Overview

The Seaglider reached the transect at 2019-03-29 04:00 UTC (fig. 1), at which point the water column was weakly stratified,
with the base of the surface mixed layer at $30\,\mathrm{m}$ and a $0.03\,°\mathrm{C}$ temperature difference across the thermocline (fig. 2 **b**).
Horizontal heterogeneity was seen across the $64\,\mathrm{km}$ transect with the eastern end persistently cooler ($0.5\,°\mathrm{C}$ to $1\,°\mathrm{C}$) than
the western end. Figure 2 **a,b** and **d** demonstrates this gradient to be non-linear and differing between the upper and lower
layers. This can also be observed in the salinity time series where a more saline region was seen between 1.8 and 2.4° E,
with fresher water at both ends of the transect. While the BML oxygen concentrations on the eastern side of the transect were
$3\,\mathrm{mmol\,m^{-3}}$ to $5\,\mathrm{mmol\,m^{-3}}$ higher than elsewhere, the saturation is similar throughout which indicates that the increased
oxygen concentration was mostly due to higher solubility at lower temperatures.

Mean wind speed was $6.7\,\mathrm{m\,s^{-1}}$, with a peak of $12.6\,\mathrm{m\,s^{-1}}$. Modelled residual surface currents were to the north-west
(counter to the typical annual residual flow), with surface waters moving into the transect area from across the bank. Surface
mean current speed was $0.17\,\mathrm{m\,s^{-1}}$ with a maximum of $0.39\,\mathrm{m\,s^{-1}}$. We calculate a cumulative total surface advection at the
centre of the transect of $52\,\mathrm{km}$ over the 27 days' occupation of the transect, with a mean residual flow of $0.02\,\mathrm{m\,s^{-1}}$. Tidal
advection was typically $5\,\mathrm{km}$ over 25 hours. The tidal elipse is shown in figure 1. Bottom water residual flow was small
($0.004\,\mathrm{m\,s^{-1}}$), with a period of flow to the north until 2019-04-12, and then to the south. As a result, bottom water is estimated
to have moved less than 9 km while the glider was on the transect. Given the 10 day transit time for the glider to return to the
same end of the transect, surface water may have been advected $19\,\mathrm{km}$ during that time.

There is disagreement regarding the importance of horizontal advection to mass balance based estimates in this region
(Queste et al., 2013; Rovelli et al., 2016; Große et al., 2016). Given the slow speed of the glider relative to the rates of change
in the inventories, and uncertainties due to the lack of observations to the north and south, it is not possible to fully resolve
any persistent or evolving horizontal $\mathrm{NO}_x^-$ (or $\mathrm{O}_2$) gradient for a 1D mass-balance. The tidally-driven advection of the
glider is largely inconsequential to the data it collects as the glider drifts with the ebb and flow of the tidal currents. Using
the modelled residual surface currents and taking the difference in near surface $\mathrm{NO}_x^-$ concentrations observed across the first
transect as indicative of the horizontal $\mathrm{NO}_x^-$ gradient ($3 \times 10^{-5}\,\mathrm{mmol\,m^{-4}}$) suggests a horizontal advective $\mathrm{NO}_x^-$ flux of
$0.05\,\mathrm{mmol\,m^{-2}\,d^{-1}}$, which can be neglected for the purposes of the present paper. The effect for $\mathrm{O}_2$ is similarly small.

The thickness of the thermocline varies substantially over the deployment ranging from $6\,\mathrm{m}$ to $53\,\mathrm{m}$ (see figure 2 **b** and
figure A1). Vertical fluxes across the thermocline are only relevant to the mass balance when the base of the thermocline is
below the $40\,\mathrm{m}$ integration depth (seen in fig. 2 **b**), and is never relevant for the full water column inventory. Diapycnal eddy
diffusivity has been shown to vary over several orders of magnitude in the North Sea (Rovelli et al., 2016). If we assume
a typical value for shelf sea diapycnal mixing of $1 \times 10^{-5}\,\mathrm{m^2\,s^{-1}}$ and use the largest observed $\mathrm{NO}_x^-$ gradient across the
thermocline of $0.77\,\mathrm{mmol\,m^{-4}}$ we estimate an upper bound for the diapycnal $\mathrm{NO}_x^-$ flux to be of the order $0.7\,\mathrm{mmol\,m^{-2}\,d^{-1}}$.
For $\mathrm{O}_2$ we see a negative flux, with a maximal gradient of $4\,\mathrm{mmol\,m^{-4}}$ seen briefly in mid April, resulting in a diapycnal $\mathrm{O}_2$
flux of the order $-4\,\mathrm{mmol\,m^{-2}\,d^{-1}}$.

**Figure 2.** Seaglider location along the transect (**a**), with associated temperature (**b**) and mixed layer depth (green line), oxygen concentration (**c**), salinity (**d**) and nitrate (**e**) profiles. Temperature, oxygen and salinity data is vertically aggregated into 3 m bins for each dive. Dark grey areas indicate data removed though quality control procedures. Samples which are excluded from the $NO_x^-$ inventory are signified with a green triangle marker.





## 3.2 $NO_x^-$ -based net community production

As we focus our analysis on the determination of new production we restrict the period of analysis to be that between peak $NO_x^-$ concentration (2019-04-06) and the time when the $NO_x^-$ inventory stops decreasing (2019-04-18). A linear least-squares fit for this time series, which integrates over three complete transect lengths, finds a linear nitrate consumption rate as shown in figure 3 and reported in table 1.

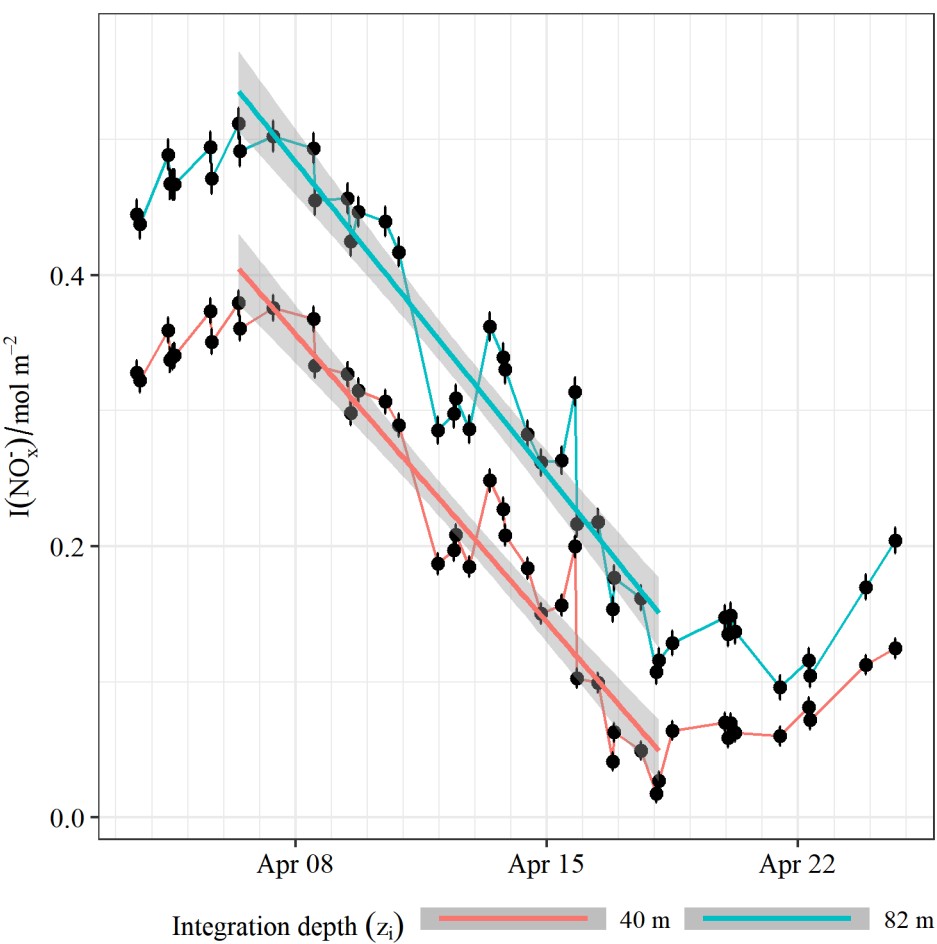

**Figure 3.** Water column integrated nitrate inventory. Points show inventory values with error bars indicating standard uncertainty. Lines indicate a linear model fit using with the grey shaded area signifying a 95 % confidence interval.

In each of these analyses the water column depth is considered to be $(82.0 \pm 0.5)\,\mathrm{m}$ based on average depth over the transect length calculated from both GEBCO bathymetry (GEBCO, 2019) and confirmed with the Seaglider altimeter. Overall the study region demonstrates $NO_x^-$ consumption rates between $0.25\,\mathrm{mmol\,m^{-3}\,d^{-1}}$ to $0.5\,\mathrm{mmol\,m^{-3}\,d^{-1}}$. We observe surface layer





(0-40 m) $NO_x^-$ consumption of $-4.5\,\mathrm{mmol\,m^{-3}}$ over 12 days with an initial inventory of $(214 \pm 7)\,\mathrm{mmol\,m^{-2}}$ drawn down

to 0 between 2019-04-06 and 2019-04-18.

### 3.3 Oxygen-based net community production

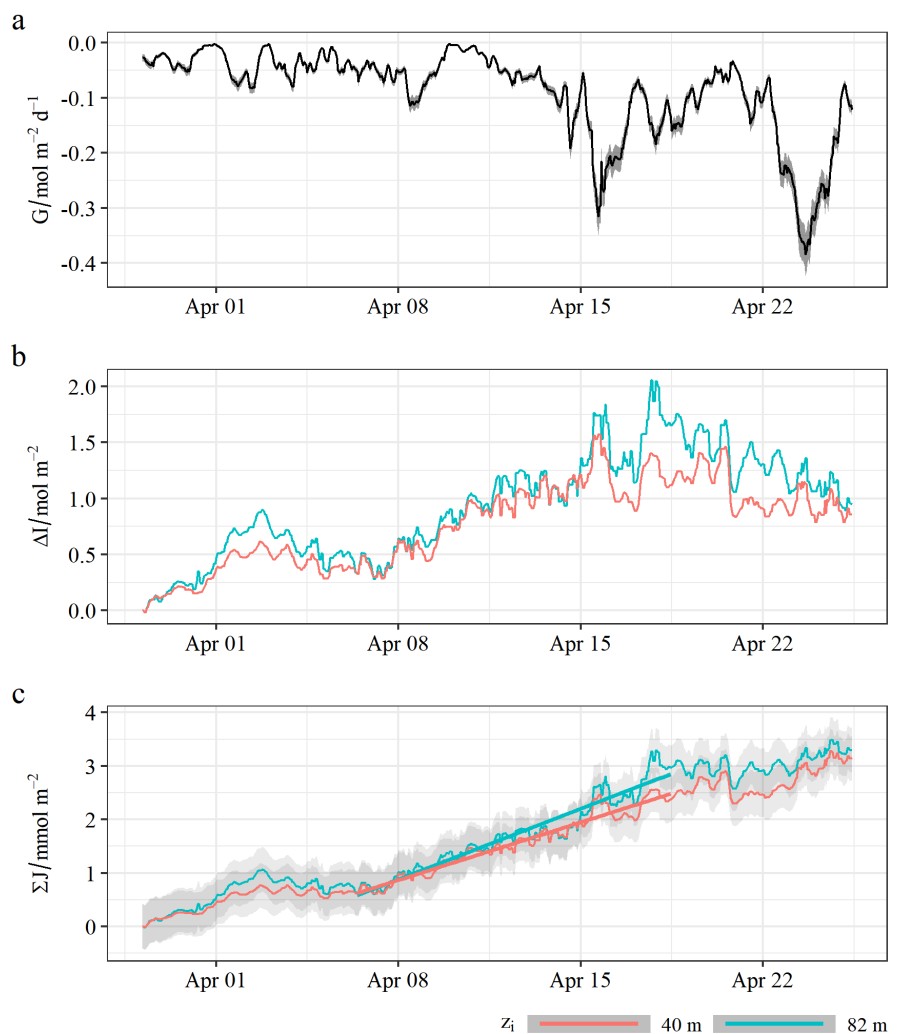

**Figure 4.** Oxygen mass balance. **a** Air-sea oxygen gas exchange with a negative sign signifying a flux out of the ocean. **b** The change in depth integrated oxygen inventory. **c** Cumulative net community production. Lines indicate linear model fit to the period associated with nitrate drawdown.

At the start of the time series the water column was slightly supersaturated with oxygen (103-104 % in the SML, and 102 % in the BML). The moderate winds associated with this period ($8\,\mathrm{m\,s^{-1}}$ to $9\,\mathrm{m\,s^{-1}}$) would be expected to induce no more





**Table 1.** Summary of $NO_x^-$ inventory changes. Inventories are calculated using either a depth consistent with the mean euphotic depth ($z_I$ = 40 m) or the full water column ($z_I$ = 82 m). $J(NO_x^-)$ is net community production in $NO_x^-$ equivalents. $J(O_2)$ is net community production in $O_2$ equivalents.

| $t_0$ | $t_1$ | $z_I$ / m | $J(NO_x^-)$ / $\mathrm{mmol\,m^{-2}\,d^{-1}}$ | $J(O_2)$ / $\mathrm{mmol\,m^{-2}\,d^{-1}}$ |
|---|---|---|---|---|
| 2019-04-06 10:00 | 2019-04-18 11:00 | 82 | $33 \pm 2$ | $188 \pm 3$ |
| 2019-04-06 10:00 | 2019-04-18 11:00 | 40 | $20 \pm 1$ | $154 \pm 2$ |

than 1 % supersaturation (Liang et al., 2013), which suggests a system which is already productive. There was some degree of
supersaturation throughout the mission both in the SML and BML, with maximum supersaturation exceeding 120 % between
2019-04-19 and 2019-04-20. The water column oxygen inventory changes are shown in figure 4 **b**. There is net outgassing
of oxygen throughout the mission (fig. 4 **a**) with cumulatively $(2.3 \pm 0.1)\,\mathrm{mol\,m^{-2}}$ released to the atmosphere over 27 days.
This outgassing is primarily driven by supersaturation through biological production, rather than due to warming of the surface
waters. Solubility changes account for only 1 % of the observed air-sea oxygen flux during this period. The water column was
net productive throughout the mission (fig. 4 **c**). Productivity $J(O_2)$ is increasing until surface $NO_x^-$ is depleted (2019-04-19)
when a marked drop in productivity followed. As such $J(O_2)$ follows a pattern consistent with the initiation of the spring
bloom followed by $NO_x^-$ limitation. The water column was likely still slightly net autotrophic at the end of April (fig. 4 **c**).
We estimate cumulative net community oxygen production over the observation period of $(3.3 \pm 0.1)\,\mathrm{mol\,m^{-2}}$ (fig. 4). As with
the $NO_x^-$ mass balance, we compare both the upper 40 m and the full water column with the results tabulated in table 1.

### 3.4 $O_2 : N$ Ratio

Comparing $J(O_2)$ and $J(NO_x^-)$ over the bloom observation period we determine the $O_2 : N$ ratios which are summarised
in table 2. We note that the uncertainty associated with these ratios is small compared to previous studies using simultaneous
fluxes owing to a more tightly constrained oxygen air-sea gas exchange flux (Alkire et al., 2012). The $O_2 : N$ ratio (table 2)
is substantially higher in the surface mixed layer relative to the rest of the water column. This is in keeping with observations
from the Alkire et al. (2012) three day North Atlantic spring bloom study which found a SML $O_2 : N$ ratio of $14.0 \pm 4.8$, and a
photic zone ($0\,\mathrm{m}$ to $53\,\mathrm{m}$) ratio of $10 \pm 6$ for the main bloom period. Our mean voluminal drawdown rates are summarised in
table 2 and are larger but broadly comparable to those of Alkire et al. (2012).

## 4 Discussion

### 4.1 Metabolic state

Our observations demonstrate a spring bloom which is highly productive and short in duration. A short duration makes it
challenging to observe with remote sensing as much of the productivity is missed entirely due to cloud cover of the study region,





typical of coastal seas at temperate latitudes. The water column is already net productive by the time the glider is occupying the transect, with weak stratification seen and partitioning of the oxygen and $NO_x^-$ across the thermocline. While the rates of change in the $NO_x^-$ concentration, relative to the speed of the glider make the determination of horizontal gradients difficult,

we can determine that in the BML $NO_x^-$ concentrations are lower towards the eastern end as seen in figure 2. Comparing those concentrations observed near $2°$ E demonstrate a decline in BML concentrations from $4\,\mathrm{mmol\,m^{-3}}$ to $3\,\mathrm{mmol\,m^{-3}}$. The deepening of the thermocline seen between 2019-04-11 to 2019-04-16 coincides with the glider moving west, and the subsequent shoaling coincides with the glider moving east. This makes it difficult to determine if this deepening is localised to the western end of the transect. However, differences in wind stress and solar heating are negligible across the transect such

that we believe the deepening occurred elsewhere along the transect simultaneously. We therefore suggest that while $NO_x^-$ continues to be consumed, the decline seen in BML $NO_x^-$ concentrations is primarily through dilution and expansion of the BML to incorporate the nutrient depleted waters from above.

### 4.2   Stoichiometry of new production

The Redfield ratio considers the remineralisation of an average marine plankton to have a 138:16 (8.6) $O_2 : N$ ratio. These ratios

are known to vary systematically across the global oceans (Li and Peng, 2002). This "traditional" ratio appears applicable to the North Atlantic, with remineralisation ratios for the deep North Atlantic yielding $8.6 \pm 1.0$ (Li and Peng, 2002).

The ratios we determine for the central North Sea bloom are lower than those observed in the North Atlantic spring bloom (table 2). $NO_x^-$ consumption appears to exceed the nitrogen requirements of the average phytoplankton given the observed oxygen production and therefore carbon fixation. However, Redfield ratios are generally not applicable for the whole water

column as the simultaneous production of organic matter together with consumption processes modifies the apparent $O_2 : N$ ratio. This is demonstrated in figure 5 where we can see that our observed $O_2 : N$ ratios could be explained by any number of simultaneous remineralisation processes. Hickman et al. (2012) found wide variations in the $C : NO_3^-$ uptake ratio associated with changes in the local light climate during summer production in the Celtic Sea. Despite these variations, the particulate organic $C : N$ ratio was observed to be close to the canonical Redfield ratio.

### 4.3   Comparison with prior regional estimates

A previous deployment of the Seaglider integrated $NO_x^-$ sensor by Vincent et al. (2018) showed a surface mixed layer (top 40 m) consumption of $-4.3\,\mathrm{mmol\,m^{-3}}$ over 21 days (2015-04-04 to 2015-04-25, $0.2\,\mathrm{mmol\,m^{-3}\,d^{-1}}$) in the central Celtic Sea during the spring bloom. The 2019 central North Sea spring bloom therefore gave $25\,\%$ to $150\,\%$ faster $NO_x^-$ drawdown rates than the 2015 central Celtic Sea bloom. Direct comparisons of the total inventory change are not possible as the Celtic Sea

Seaglider did not sample the stratifying surface waters for $NO_x^-$ during the onset of the 2015 bloom.

The spring bloom in the seasonally stratified regions of the North Sea is thought to represent 40 % of the typical regional annual production (Richardson and Pedersen, 1998). Several previous studies have derived local production estimates for the region north of the Dogger Bank, but these have focused on the summer months (Weston et al., 2005; Bozec et al., 2005). There are relatively few contemporary NCP observations for this region and most previous studies aggregate their produc-





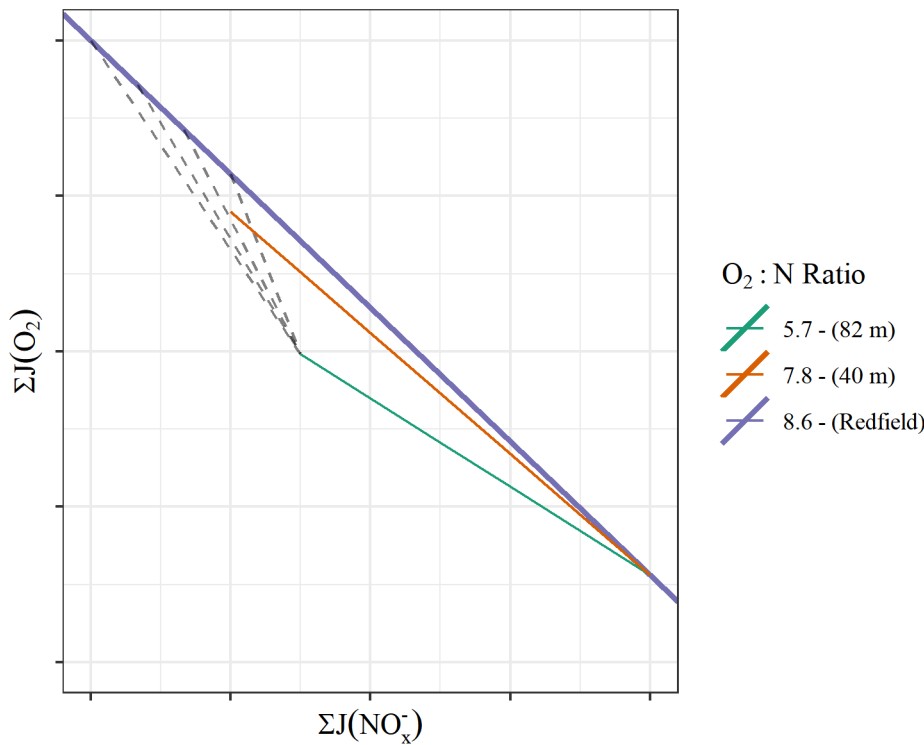

**Figure 5.** Conceptual diagram which shows how the observed $O_2 : N$ ratios (shown with coloured lines) can be explained by the presence of any number of simultaneous remineralisation processes (dashed lines).

**Table 2.** Voluminal comparison with recent studies

|  | $z_I$ / m | $J(O_2)$ / mmol m$^{-3}$ d$^{-1}$ | $J(NO_x{}^-)$ / mmol m$^{-3}$ d$^{-1}$ | $O_2 : N$ ratio |
|---|---|---|---|---|
| This Study - Central North Sea (Spring) | 0 to 40 | $3.86 \pm 0.05$ | $0.53 \pm 0.05$ | $7.8 \pm 0.5$ |
|  | 0 to 82 | $2.29 \pm 0.04$ | $0.45 \pm 0.05$ | $5.7 \pm 0.4$ |
| Vincent et al. (2018) - Celtic Sea (Spring) | 0 to 40 |  | 0.2 |  |
| Alkire et al. (2012) - North Atlantic (Spring) | 0 to 53 | $2.9 \pm 0.8$ | $0.3 \pm 0.1$ | $10 \pm 6$ |
| Weston et al. (2005) - Central North Sea (Summer) |  |  | 0.72 |  |





tivity estimates over very large spatial or temporal scales. Such a broad approach provides limited information to describe
productivity at the spatial and temporal scales comparable to modern biogeochemical models which operate on $<10\,\mathrm{km}$ resolution. Where disparities arise between model and observation, it is is difficult to determine which regions or processes are
inadequately captured. Therefore, accurate advice from the ocean observing community on how models might be improved is
heavily constrained.

Richardson and Pedersen (1998) determined the winter nitrate maximum in the central stratified North Sea to be $8\,\mathrm{mmol\,m^{-3}}$
based on 1993 OSPAR data and the 2017 Interim OSPAR Report determined it to be $6\,\mathrm{mmol\,m^{-3}}$ to $8\,\mathrm{mmol\,m^{-3}}$ between
2006 and 2014. The deployment of a similar glider also fitted with a $\mathrm{NO_x^-}$ analyser between November 2017 and January 2018
observed a maximum of $(7.0 \pm 0.2)\,\mathrm{mmol\,m^{-3}}$ (Birchill et al., 2021). Using $8\,\mathrm{mmol\,m^{-3}}$ as the winter nutrient concentration,
taking the typical thermocline depth in a given year to be $30\,\mathrm{m}$ with zero vertical mixing and a C:N ratio of 5.68 they calculated

total annual new production for the stratified North Sea to be $(1360 \pm 680)\,\mathrm{mmol\,m^{-2}}$ (C). By contrast our $\mathrm{NO_x^-}$ inventories,
with a lower winter nitrate maximum, indicate a total spring bloom new production of $(1043 \pm 122)\,\mathrm{mmol\,m^{-2}}$ (C) using the
above C:N ratio.

Bozec et al. (2006) calculated dissolved organic carbon-based NCP for the International Council for the Exploration of
the Sea (ICES) regions which contain the AlterEco transect (#4 & #14). For comparison we convert these values of NCP to

oxygen equivalents using an $\mathrm{O_2}$ :C ratio of 1.4. This conversion provides a surface mixed layer NCP of $0.93\,\mathrm{mmol\,m^{-3}\,d^{-1}}$
and a bottom water average NCP of $-0.14\,\mathrm{mmol\,m^{-3}\,d^{-1}}$ for April (Bozec et al. (2006) do not provide uncertainties). The
large temporal and spatial integration range likely fails to resolve much of the local short term productivity.

### 4.4    Interannual variability

Near surface $\mathrm{NO_x^-}$ and mid-water oxygen from the 2007-2008 Cefas SmartBuoy mooring (fig. 1) are shown in figure 6

together with the comparable Seaglider observations from 2019. We see peak $\mathrm{NO_x^-}$ concentrations prior to bloom onset are
consistent $(7.0 \pm 0.5)\,\mathrm{mmol\,m^{-3}}$ across all observed years within this region. Winter $\mathrm{NO_x^-}$ concentrations are predicted to
reduce in the North Sea due to reduced exchange with the Atlantic Ocean (Gröger et al., 2013), but we do not see any evidence
for this in our data. A more complete knowledge of the water column vertical structure is required to accurately determine
NCP rates from these observations. Extrapolating observations at the surface buoy to an unknown mixed depth to determine an

inventory introduces substantial uncertainties. We can use a subset of glider observations to demonstrate this; using only the
near surface value to calculate a $40\,\mathrm{m}$ inventory results in a 23 % larger calculated $\mathrm{NO_x^-}$ drawdown. However, the trend in
figure 6 suggests similar maximal rates of $\mathrm{NO_x^-}$ consumption.

The duration and intensity of the $\mathrm{NO_x^-}$ drawdown demonstrates marked interannual variability. While there is an expected
inverse correlation between pre-bloom temperature and oxygen concentration, oxygen saturation does not appear to be solely

controlled by differing sea temperatures or bubble injection as 2007 is under-saturated in mid-March (93 %), 2008 is very
close to saturation (99 %) and 2019 2-3 % supersaturated. We note that while there is a general trend of increasing sea surface
temperatures in the North Sea (Capuzzo et al., 2018) 2019 was cooler then 2007. Große et al. (2016) suggested that year-to-
year variations in the oxygen concentrations are mostly caused by variations in local primary production. Warmer years are



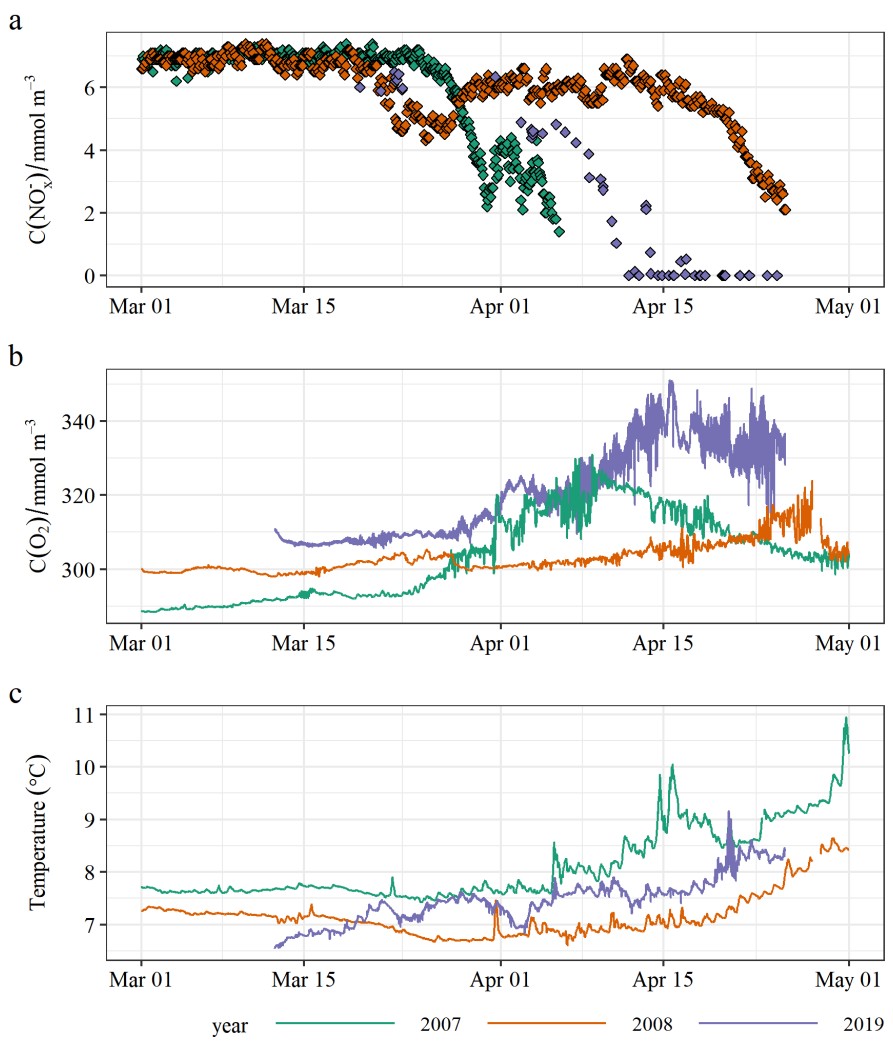

**Figure 6.** $NO_x^-$ and oxygen concentration observations from the 2007-2008 North Dogger Cefas SmartBuoy mooring together with the 2019 AlterEco Seaglider (this study). $NO_x^-$ and temperature observations are from 1 m depth, while oxygen concentrations are from 25 m in 2007 and 35 m in 2018, for 2019 all data between these depths is shown.

thought to induce an earlier spring bloom (Capuzzo et al., 2018). This is reflected in these data, with late March near surface
sea temperatures correlating with the observed onset of the bloom.

## 5 Conclusions

We present well resolved productivity estimates using two complementary methods from the central North Sea using a single
autonomous vehicle. These results represent the most complete examination of the spring bloom formation within this region
and demonstrate its highly episodic nature. A feature which is typically hard to observe with typical ship surveys or remote
sensing. These observations highlight the strength of autonomous vehicles as a key part of future monitoring strategies, com-
plementing remote sensing and biogeochemical modelling efforts. By comparing the rates of NCP through the two methods
we determine the $O_2 : N$ ratio of new production within the water column, finding $O_2 : N$ ratios in the near surface close to the
canonical Redfield North Atlantic ratio, while remineralisation processes in the deeper parts of the water column tend to lower
the apparent ratio.

*Author contributions.* Tom performed the NCP analysis and prepared the original draft. Tom, Jan, Antony and Naomi provided the data
curation and validation. Tom, Antony, Alex and Matthew conducted the glider fieldwork. All authors contributed to experimental design and
writing of the manuscript

*Competing interests.* Alex is a co-founder and employee of Clearwater Sensors. The remaining authors declare this research was conducted
in the absence of any potential conflict of interest.

*Acknowledgements.* This work was funded by the UK National Environment Research Council (NERC), the UK government's Department
for Environment, Food and Rural Affairs (Defra), the World Wide Fund for Nature (WWF). NERC/DEFRA grant numbers NE/P013899/1,
NE/P013902/2, NE/P013740/1 and NE/P013864/1. The North Dogger SmartBuoy was funded by Defra as part of the ME3205 and SLA25
projects.

*Data availability.* The Seaglider data is available at the British Oceanographic Data Centre (BODC) (https://www.bodc.ac.uk/). See also
Birchill et al. (2021) for a more thorough description. SmartBuoy data is available at the Cefas Data Hub (https://data.cefas.co.uk/).

*Code and data availability.* R and STAN code used for this analysis is available by request





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

## Appendix A: Supplementary Material

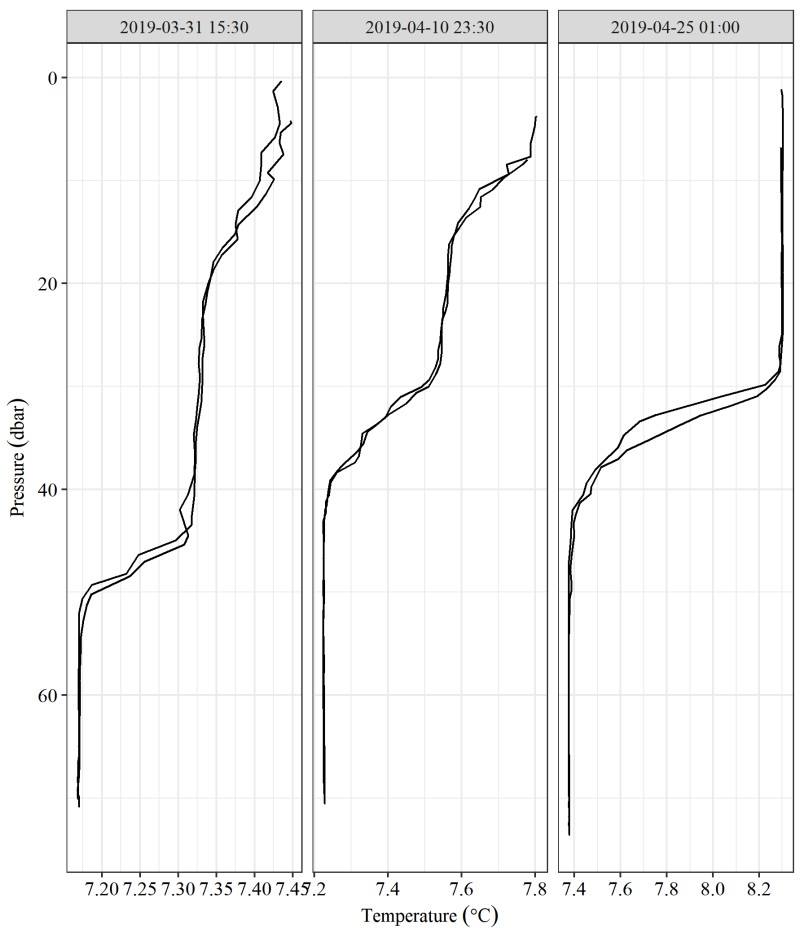

**Figure A1.** Example vertical temperature profiles recorded by the Seaglider.





**Table A1.** Summary of $NO_x^-$ calibration standards ($\mathrm{mmol\,m^{-3}}$).

| Standard | Certified value | Determined value |
|----------|-----------------|------------------|
| BZ0038 | $44.410 \pm 0.340$ | $45.28 \pm 0.15$ (n=2) |
| CD1515 | $5.630 \pm 0.050$ | $5.58 \pm 0.03$ (n=4) |

## A1 Oxygen NCP model

We can describe the evolution of oxygen over time as the sum of various fluxes:

$$z_I \frac{\mathrm{d}C_I}{\mathrm{d}t} = G + M + E - A + J \tag{A1}$$

where $C_I$ is the average oxygen concentration within integration depth $z_I$, $G$ is the air-sea gas exchange with a flux into the ocean being defined as positive, $M$ is mixing comprising horizontal ($M_A$) and vertical ($M_z$) mixing, $E$ is entrainment, $A$ is horizontal advection and $J$ is net community production.

Air-sea gas exchange is calculated as

$$G = k_w \left( C_{\mathrm{sat}}(1+B)\frac{P}{P_0} - C_s \right) \tag{A2}$$

Where $k_w$ is the air-sea transfer velocity parameterisation of Nightingale et al. (2000) derived using the `airsea` R library (Hull and Johnson, 2015), using the Schmidt number parameterisation of Johnson (2010). $C_s$ is the near surface oxygen concentration. $C_{\mathrm{sat}}$ is the concentration of oxygen if it were in equilibrium with the atmosphere at a pressure of 1013.25 hPa

(Garcia and Gordon (1992), using the fit to the Benson and Krause data). $P$ is local air pressure and $P_0$ standard atmospheric pressure (1013.25 hPa). $B$ is the wind speed and temperature-dependent bubble supersaturation term from Liang et al. (2013). This bubble term has a relatively large effect on net community production estimates when $C_s$ is close to $C_{\mathrm{sat}}$ (Emerson and Bushinsky (2016), Hull et al. (2016), Liang et al. (2017)). Gas exchange is only relevant within the ventilated partition of the water column, which is equivalent to the actively mixed surface layer (known as mixing layer (Brainerd and Gregg, 1995))

($z_{\mathrm{mix}}$). If $z_I < z_{\mathrm{mix}}$ then $G$ needs to be scaled by the factor $z_I/z_{\mathrm{mix}}$.

Horizontal advection can be described by equation A3.

$$A = \left( u\frac{\mathrm{d}C}{\mathrm{d}x} + v\frac{\mathrm{d}C}{\mathrm{d}v} \right) z_i \tag{A3}$$

where $\frac{\mathrm{d}C}{\mathrm{d}x}$ is the zonal oxygen gradient and $u$ is the zonal velocity, with equivalent terms for the meridional ($v$) direction.

We can not determine $A$ from our glider data, as gliders are semi-Lagrangian they are displaced by tidal currents and as such

the method of Hull et al. (2020) is not applicable. Gliders are also relativity slow compared with the observed rates of oxygen concentration change and the residual currents along the transect.

Depending on the choice of integration depth i.e. the volume of water we calculate NCP over, the fluxes to consider change. For example Barone et al. (2019) eliminated the influence of entrainment by defining their integration depth corresponding





to an isopycnal, which is always deeper than the base of the surface mixed layer. Often the integration is restricted to the

productive part of the water column (euphotic depth) e.g. Bushinsky and Emerson (2015) or Binetti et al. (2020). However, the shallow nature of the study area presents an opportunity to reduce the overall uncertainty by calculating the change oxygen inventory over the whole water column, the depth of which is known with a high level of certainty. There is little variation in depth across the E-W transect and as such we calculate our full water column mass balance in terms of the average total water column depth ($z_i = (82 \pm 0.5)$ m).

$z_{\mathrm{mix}}$ is not necessarily equivalent to the thermocline depth, as diurnal warming can reduce the ventilated water column to a region of water above the persistent seasonal thermocline, and it is this volume which is influenced by air-sea gas exchange directly. Typically the dynamic nature of this layer, and the difficulty determining its extent from density profiles can introduce significant uncertainties to the magnitude of the air-sea gas flux. The 0.2 °C threshold we use to determine the SML visually agrees well with the oxygen profiles used for determining the value of $C_s$.

By integrating either the entire water column, or to a depth which encompasses the BML, our mass balance differential equation is reduced to equation A4: We simply need to estimate the oxygen concentration within the mixing layer for use in determining the gas exchange flux. This also means entrainment and diapycnal mixing fluxes can be ignored as these do not influence the full water column inventory. Rovelli et al. (2016) observed that within the central North Sea these vertical fluxes can be substantial, while being both difficult to quantify and temporally variable. Formulating the mass balance to avoid the

need for these terms is thus highly advantageous.

$$\frac{\mathrm{d}I}{\mathrm{d}t} = G + J \tag{A4}$$

In order to best include our systematic and measurement uncertainties we adopt a Bayesian Markov-Chain Monte-Carlo approach to infer our parameter estimates. We implement the probabilistic model described in equation A5, where theˆoperator signifies our observations of the unknown (latent) state parameter. $C_s$ is the concentration of oxygen in the surface actively

mixed layer ($z < z_{\mathrm{mix}}$). $I$ is the oxygen concentration integrated over our integration depth ($z_I$). $\varepsilon_C s$ is the uncertainty bias in the oxygen measurements, for this study considered to be $\pm 1\,\mathrm{mmol\,m^{-3}}$. This model equates to a 15 % error on $k_w$, a 50 % error on $B$. $\varepsilon_I$ is calculated as the standard error of $I$. $\alpha$, $\beta$ and $\sigma$ are left as improper priors, as in a standard least squares regression. Implementation of this model is using the probabilistic programming language Stan (Carpenter et al., 2017). Output variables are presented with 95 % credible intervals, which are the interval within an unobserved parameter value falls with a

particular probability. These are therefore thus analogous to frequentist confidence intervals.





$$\Sigma J \sim \mathcal{N}(\alpha + \beta t, \sigma)$$

$$\Sigma J_t = (\Sigma I_t - I_0) - (\Sigma G_t \Delta t)$$

$$G = k_{\mathrm{w}} \left( C_{\mathrm{sat}}(1 + B) \frac{P}{P_{\mathrm{a}}} - C_{\mathrm{s}} \right)$$

$$\hat{I} \sim \mathcal{N}(I, \varepsilon_I)$$

$$\hat{k_{\mathrm{w}}} \sim \mathcal{N}(k_{\mathrm{w}}, \hat{k_{\mathrm{w}}} 0.15)$$

$$\hat{C}_s \sim \mathcal{N}(C_s, \varepsilon_{Cs})$$

$$\hat{B} \sim \mathcal{N}(B, \hat{B} 0.5)$$

$$\hat{C_{\mathrm{sat}}} \sim \mathcal{N}(C_{\mathrm{sat}}, 1) \tag{A5}$$

## A2 SmartBuoy Mooring

The SmartBuoy monitoring station known as "North Dogger" was occupied between 2007-02-25 and 2008-09-14. The moorings consisted of a Cefas ESM2 data logger with an Aanderaa 3835 oxygen optode, Aanderaa 3319B (conductivity and temperature) and a Envirotech NAS-3X nutrient analyser. The NAS-3X performs the same wet-chemistry method as the LoC.