# Peer review of "Simultaneous assessment of oxygen and nitrate-based net community production in a temperate shelf sea from a single ocean glider"

_Biogeosciences, 2021_

## Author Response (AR1)

**BG-2021-170 Response to review**

We thank both of our referees for reviewing our manuscript and for providing helpful comments. We feel we have addressed all of their points and the manuscript is much improved as a result.

**Response to review 1**

**Detailed comments**

1. Ken Johnson's and related works typically have used UV (SUNA) sensors in high nitrate regions, these are are inappropriate for the central North Sea given the very low concentrations seen in summer, AlterEco is the second deployment of the microfluidic sensor and the first time bloom production rates have been observed with a glider. We'll adjust the text to express this.

   - Indeed, the larger requirements of the AlterEco study precluded a transect optimised for NCP determination. A review of optimal sampling strategy is perhaps beyond the scope of this paper however.

2. We believe the novel and useful aspect of this work is in the concomitant determination of NCP through the two methods and therefore the determination of the stoichiometry. The carbon cycling, Dogger bank ecosystem description and deoxygenation sections are included to indicate why determining NCP as a ecosystem parameter is important. However, in retrospect we agree that these sections are longer than needed and we'll revise accordingly.

3. as above

4. Figure 2 simply shows the observed variables in unit natural to the sensor observations, which are also adhere to the NetCDF Climate and Forecast standard names list. In table 1, Columns 4 and 5 do already state O2 based NCP and NOx based NCP in depth-integrate units. We avoided using N for NCP as we use statistical notation for the model later which uses N() as the Gaussian distribution. J was used by Steve Emerson in 1987, and many related works since and we mostly followed with that notation. As referee 2 has noted some clarification of the J term is required so we've amended the manuscript.

5. The term "ecosystem services" was first coined in 2007 and has found widespread use. There are now several hundred published papers in biogeosciences which include the phase "ecosystem services" in the abstract, We think it's a succinct way of talking about the value of natural systems.

6. as with 2. and 3. We'll change the paper to make the focus clear.

7. $Z_{\mathrm{mix}}$ is still required for determining the value of $C_s$ used for the air-sea gas exchange calculation so was included for completeness, this is in the supplementary material. We'll move these lines there to improve the flow of the methods section.

8. The tidal ellipse is centred in the middle of the transect and scaled to be visible. We refer to the historic buoy data when discussing interannual variability.

9. No, this Seaglider was not fitted with Wetlabs or similar chlorophyll fluorometer.

10. We're using the buoy observations to discuss if there is evidence for changes in the nutrient dynamics in this region over the last several years as has been suggested by modeling studies. The new observations from the glider and the previously unpublished observations from the buoy are useful in this regard and we feel are deserving of the readers' attention.

**Response to review 2**

- We have amended both the main text and the appendix to provide a more clear description of the J term and how it is used elsewhere in the paper.

- We have corrected line 234, and importantly figure 3 which had been generated with the incorrect subset of data for 40 m. This line was misleading because we use the slope of the regression for the rate rather than just using the start and end points.

**Detailed comments**

- Agreed, total oxidised nitrogen is a bit verbose while nitrogen on it's own suggests $N_2$. We have included a reference to the LoC in the abstract which should make it clear our observations are NOx.

- Line 153, We use the pressure, we've made this clear in the text.

- Line 163, "a" dropped

- Line 205, We've added "not to scale" to the caption, it's too small to see easily when drawn 1:1.

- Line 231, corrected

- Line 233, figure 3 has now been corrected as noted above. The y axis is mol m**-2**, (now changed to mmol), so there is no need to scale by the water column depth.

- Line 234, as noted above

- Line 244, moving from 7 to 8 degrees causes around a 2% change in the solubility concentration, but not necessarily in the overall oxygen flux.

- We have changed the use of productivity to production throughout

- Line 263, corrected

- Line 327, corrected

- Line 536, corrected

- Figure 1, Axis labels added and text made larger

- Figure 5, We have removed the numbers from the slopes and the gridlines, we're keen to keep this looking conceptual.